# Analysis of the Spatiotemporal Evolution of the Net Carbon Sink Efficiency and Its Influencing Factors at the City Level in Three Major Urban Agglomerations in China

**DOI:** 10.3390/ijerph20021166

**Published:** 2023-01-09

**Authors:** Shiguang Shen, Chengcheng Wu, Zhenyu Gai, Chenjing Fan

**Affiliations:** College of Landscape Architecture, Nanjing Forestry University, Nanjing 210037, China

**Keywords:** urban agglomerations, carbon budget, NCSE, influencing factors

## Abstract

The implementation of carbon peaking and carbon neutrality is an essential measure to reduce greenhouse gas emissions and actively respond to climate change. The net carbon sink efficiency (NCSE), as an effective tool to measure the carbon budget capacity, is important in guiding the carbon emission reduction among cities and the maintenance of sustainable economic development. In this paper, NCSE values are used as a measure of the carbon budget capacity to measure the spatiotemporal evolution of the carbon neutral capacity of three major urban agglomerations (UAs) in China during 2007–2019. The clustering characteristics of the NCSE of these three major UAs, and various influencing factors such as carbon emissions, are analyzed using a spatiotemporal cube model and spatial and temporal series clustering. The results reveal the following. (1) From the overall perspective, the carbon emissions of the three major UAs mostly exhibited a fluctuating increasing trend and a general deficit during the study period. Moreover, the carbon sequestration showed a slightly decreasing trend, but not much fluctuation in general. (2) From the perspective of UAs, the cities in the Beijing–Tianjin–Hebei UA are dominated by low–low clustering in space and time; this clustering pattern is mainly concentrated in Beijing, Xingtai, Handan, and Langfang. The NCSE values in the Yangtze River Delta UA centered on Shanghai, Nanjing, and the surrounding cities exhibited high–high clustering in 2019, while Changzhou, Ningbo, and the surrounding cities exhibited low–high clustering. The NCSE values of the remaining cities in the Pearl River Delta UA, namely Guangzhou, Shenzhen, and Zhuhai, exhibited multi-cluster patterns that were not spatially and temporally significant, and the spatiotemporal clusters were found to be scattered. (3) In terms of the influencing factors, the NCSE of the Beijing–Tianjin–Hebei UA was found to be significantly influenced by the industrial structure and GDP per capita, that of the Yangtze River Delta UA was found to be significantly influenced by the industrial structure, and that of the Pearl River Delta UA was found to be significantly influenced by the population density and technology level. These findings can provide a reference and suggestions for the governments of different UAs to formulate differentiated carbon-neutral policies.

## 1. Introduction

Global carbon dioxide emissions reached an unprecedented peak in 2020 [1], and the resulting climate change and greenhouse effect have attracted widespread attention. Determining how to cope with and promote the early achievement of the dual carbon goals in each country has become an important issue for global academics [2]. The reduction of greenhouse gas emissions and the improvement of carbon sequestration in terrestrial ecosystems are two key ways to achieve a carbon balance [3], as well as effective measures by which to achieve carbon neutrality in cities [4]. In terms of carbon emission reduction, the high speed of urbanization in developing countries has driven economic development, and cities account for more than 70% of global CO_2_ emissions [5] and are under tremendous pressure. However, due to the ambiguous understanding of the relationship between emission reduction and economic development, many regions try to reduce emissions by controlling the rate of industrial development; this leads to a contradiction between economic development and the reduction of CO_2_ emissions. In terms of carbon sinks, the impact of land use and land cover changes (LUCCs) on the amount of carbon sequestered by regional ecosystems has been considered, as agroforestry ecosystems can increase natural carbon sinks, with positive effects on the carbon balance [6,7]. China, as the world’s largest developing country, has actively taken relevant measures and proposed the dual carbon goal of reaching peak carbon emissions by 2030 and achieving carbon neutrality by 2060 [8].

For developing countries, limiting economic development is an unsustainable path to carbon-neutral development. The improvement of the carbon emission efficiency (CEE) should not only avoid affecting the economy, but should also consider the role of carbon sequestration by vegetation (CSV). From the perspective of developing countries, the CEE has been calculated by taking the gross domestic product (GDP) as the desired output and CO_2_ emissions as the undesired output with constant labor, capital, and energy inputs; however, this process does not consider the role of CSV in promoting the urban carbon balance [9,10]. The sole reliance on reducing carbon emissions can affect economic development, especially for developing countries. The delineation of China’s three zones and three lines and the long-term implementation of afforestation programs have led to CSV playing a substantial role as a natural carbon sequestration process. Research has found that from 2020 to 2050, China’s forest vegetation will absorb 22.14% of CO_2_ emissions [11]. To integrate the CEE and CSV, the net carbon sink efficiency (NCSE) is used as an indicator to measure the carbon balance standard. The concept of NCSE is the efficiency value calculated by adding CSV as the desired output to the evaluation framework of the CEE. It is sustainable for economic development, more in line with the national conditions of developing countries, and also aids in the understanding of the potential carbon reduction under different levels of economic development.

Current research on CEE is usually focused on two topics, namely computational methods and impact factor analysis. Among the different computational methods of CEE, the most widely used is data envelopment analysis (DEA), and several extension models have become the mainstream methods for the calculation of environmental efficiency [12]. Based on its measurement, scholars have used panel models and spatial econometric models to study the impacts of various factors on regional CEE. It has been found that technological progress, economic development, population agglomeration, foreign investment, and the industrial structure all have significant impacts on CEE. Technological aspects are crucial for the reduction of pollution emissions [13], and green technology is often considered a key tool to achieve energy efficiency, emission reduction, and economic growth as an influential factor to improve CEE [14]. Economic development, population growth, and rapid urbanization accelerate CO_2_ emissions [15,16]. Factors related to urban expansion [17] and overpopulation [18] cause LUCCs, which increase the impervious surface [19] and hinder the carbon uptake of soils to impact the carbon balance of terrestrial ecosystems, which in turn has a negative impact on regional CEE [20]. Foreign investment also plays a significant role in promoting energy efficiency; the improvement of foreign investment quality can promote energy efficiency and a low-carbon economy [21]. Industrial restructuring is the main way to reduce the energy intensity [22] and CO_2_ emissions [23,24] and is also one of the most important means by which to promote CEE improvement [25].

The studies reviewed previously are deficient in at least two aspects. On the one hand, the contemporary studies on carbon balance mostly analyze cross-sectional data from some specific years and cannot unify both continuous time series and spatial dimensions, inevitably ignoring some periodically occurring spatiotemporal clustering patterns [26]. The use of a spatiotemporal cube model (STC) combined with local outliers may allow for the analysis of the distribution characteristics of NCSE values both spatially and temporally [27]. On the other hand, most of these previous studies were focused on the national scale [13,28] and analyzed the different characteristics of CEE present in the spatial pattern. At the city level, research on the relationship between LUCCs and the carbon balance has found that the overall carbon balance level of most cities is in a deficit [29,30,31,32,33,34], and a carbon balance cannot be achieved within the urban system. However, the scale involved in national-level studies is too wide, and it not only ignores the dynamic characteristics of urban development but also makes it difficult to reveal the microscopic changes in land use and the differential effects of carbon emissions on NSCE. Moreover, the characteristic scale of city-level studies is too small, and there may be an overlap of administrative boundaries regarding the calculation of carbon emissions and sequestration, which will also produce neighbor-avoidance effects on neighboring cities. In contrast, the scale of city agglomerations, as aggregates of neighboring spatial and natural features, can be better used to study the LUCCs brought about by urban expansion and the impacts of carbon revenues and expenditures on NSCE. It can also be used to formulate reasonable regional development measures from the perspective of regional integration. However, there are relatively few research results at this scale.

In general, research on urban agglomerations (UAs) over long time series can help to reveal the key factors of NCSE differences among UAs. Beijing–Tianjin–Hebei (BTH), the Yangtze River Delta (YRD), and the Pearl River Delta (PRD), as the three most typical UAs in China, have important impacts on carbon emissions in China and even globally. Therefore, in this study, these three major UAs were selected as the research object, the temporal evolution and spatial distribution characteristics of CO_2_ emissions were analyzed, and the CSV and NCSE of the UAs during 2007–2019 were investigated by the STC model. A comparative analysis of the influencing factors of the NCSE was conducted based on the two-way fixed-effects model to explore the commonality and individuality of carbon revenues and expenditures among the three UAs.

## 2. Data Acquisition and Research Methodology

### 2.1. Overview of the Study Area

The BTH, YRD, and PRD UAs were selected as the research objects (Figure 1), among which the BTH UA includes 13 cities in the Beijing, Tianjin, and Hebei Provinces, with a total area of 218,000 km^2^, including three major geographical units: the northwest mountainous area, the southeast plain, and the eastern Bohai sea. The core area of the YRD UA includes 16 cities in Jiangsu Province, Zhejiang Province, and Shanghai City, with a total area of 131,000 km^2^. The cities are located in the lower reaches of the Yangtze River bordering the Yellow Sea and the East China Sea, and at the meeting place of the rivers and the sea. The region is characterized by many coastal ports along the river and the alluvial plain formed before the Yangtze River entered the sea. The PRD UA includes Guangzhou, Foshan, Zhaoqing, Shenzhen, Dongguan, Huizhou, Zhuhai, Zhongshan, Jiangmen, and nine other cities, with a total area of 55,500 km^2^. The cities are located in the south–central part of China’s Guangdong Province at the mouth of the Pearl River. The three major UAs account for around 5% of the country’s total land area and 33% of the population, but contribute nearly 44% of the country’s GDP [35]. The three major UAs are important regions for China’s economic development, and the study of the influencing factors of their NCSE can provide an important reference for China to formulate a dual carbon policy.

### 2.2. Data Sources and Processing

#### 2.2.1. Calculation of the Main Variables

(1)Calculation of carbon emissions and carbon sequestration

The carbon emission factor method published by the Intergovernmental Panel on Climate Change (IPCC) is the most widely used to calculate the amount of carbon released [36,37]. However, the default emission factor of the IPCC is not fully applicable to the national conditions of China. The carbon emission accounting data used in this study were derived from the China Carbon Accounting Database of the Carbon Emission Accounts and Datasets (CEADs) [38]. This database compares the Chinese default factors issued by the IPCC and the National Development and Reform Commission of China (NDRC), based on which a new carbon emission factor is developed based on the actual situation of each industry [39]. The data of various types of carbon emissions reported in this database were summed to obtain the total carbon emissions data, and the specific expression is given by the following equation:(1)$$CE_{\mathrm{it}}=CEE_{it}+CEA_{it}+CEI_{it}+CEU_{it}$$
where *CE* is the carbon emissions data, *CEE* is the carbon emissions due to energy consumption, *CEA* is the carbon emissions due to agricultural production, *CEI* is the carbon emissions due to industrial production, and *CEU* is the carbon emissions due to urban waste. The subscripts *i* and *t* denote the city and year, respectively.

The carbon sequestration factor method is used for the calculation of the solid carbon data [40,41]. LUCC data were obtained from Landsat’s annual Chinese land cover dataset [42]. The various types of LUCC data were extracted and fused by municipal division, and the areas of different LUCC types were calculated separately for each city. Carbon sequestration was estimated separately for different LUCC types, and the sequestration coefficients of specific LUCC types are reported in Table 1. Among them, cropland is both a carbon source and a carbon sink, and the carbon sequestered by cropland mainly comes from cropland soil. The specific calculation method for carbon sequestration data is given by Equation (2).
(2)$$CS_{\mathrm{it}}=\sum T_{it}\cdot \delta_{}$$
where *CS* is carbon sequestered the amount contributed by different LUCC types, *T* is the area of each LUCC type, and δ is the carbon sequestration coefficient of each LUCC type. The subscripts *i* and *t* denote the city and year, respectively, and the carbon sequestration coefficients for different utilization types are reported in Table 1.

(2)NCSE calculation

The NCSE value is calculated by adding CSV as a desired output to the CEE assessment framework to calculate the efficiency value [48] as a measure of carbon balance, namely carbon neutrality without affecting economic development. In this study, the DEA approach was used to measure the urban NCSE as a measure of carbon neutrality. Traditional DEA models, such as the Charnes, Cooper, and Rhodes (CCR) and Banker, Charnes, and Cooper (BCC) models, calculate the production efficiency when only the desired output is available. The super-efficient slacks-based measure (SBM) model, however, is based on the DEA model. It overcomes the shortcomings of the ordinary DEA model, namely that it cannot evaluate multiple decision-making units (DMUs). It distinguishes the differences between the effective units of DEA, and it can rank the DMUs effectively. Thus, the decision results will be more realistic [49]. In this case, it was assumed that there are *n* DMUs, each of which has three vectors, namely inputs, desired outputs, and non-desired outputs. The expected output of S1 and the undesired output of S2 were generated using *m* inputs, respectively, represented by *x* ∈ *R^m^*, *y^g^* ∈ *R^S1^*, and *y^b^* ∈ *R^S2^*. The matrices *X*, *Y^g^*, and *Y^b^* are, respectively, defined as *X* = [*x*_1_, *x*_2_, …, *x_n_*] ∈ *R^m×n^*, *Y^g^* = [$y_{1}^{g},y_{2}^{g},\ldots ,y_{n}^{g}$] ∈ *R^S*1*×n^*, and *Y^b^* = [$y_{1}^{b},y_{2}^{b},\ldots ,y_{n}^{b}$] ∈ *R^S*2*×n^*. Assuming that *X* > 0, *Y^g^* > 0, and *Y^b^* > 0, the production possibility set is defined as follows [13].
(3)$$P=\{(x,y^{g},y^{b})/x\geq X\theta ,y^{g}\geq Y^{g}\theta ,y^{b}\leq Y^{b}\theta ,\theta \geq 0\}$$

This equation incorporates the non-desired output into the SBM model of the evaluation DMU ($x_{0}$, $y_{0}^{g}$, $y_{0}^{b}$), as follows:(4)$$\rho =\min\frac{1-\frac{1}{m}\sum_{i=1}^{m}\frac{S_{i}^{-}}{x_{i0}}}{1+\frac{1}{S_{1}+S_{2}}\left(\sum_{r=1}^{S_{1}}\frac{S_{r}^{g}}{y_{r0}^{g}}+\sum_{r=1}^{S_{2}}\frac{S_{r}^{b}}{y_{r0}^{b}}\right)}$$
$$s.t.:x_{0}=X\theta +S^{-},\;y_{0}^{g}=Y^{g}\theta -S^{g},\;y_{0}^{b}=Y^{b}\theta -S^{b};\;S^{-}\geq 0,\;S^{g}\geq 0,\;S^{b}\geq 0,\;\theta \geq 0.$$
where $S_{i}^{-}$ denotes the input redundancy, $S_{r}^{g}$ denotes the desired output deficiency, $S_{r}^{b}$ denotes the desired output excess, θ denotes the weight vector, and ρ denotes the efficiency value, which is within the range of [0,1]. The DMU is determined to be valid only when ρ = 1; when 0 < ρ < 1, the unit is inefficient. In the efficiency evaluation, the DMU will mostly appear in the 100% efficiency state, so it is necessary to distinguish the DMU and the influencing factors to ensure that the efficiency value is close to the real level. The model for this is
(5)$$\rho^{*}=\min\frac{\frac{1}{m}\sum_{i=1}^{m}\frac{{\bar{x}}_{i}}{x_{i0}}}{\frac{1}{S_{1}+S_{2}}(\sum_{r=1}^{S_{1}}\frac{{\bar{y}}_{r}^{g}}{y_{r0}^{g}}+\sum_{r=1}^{S_{2}}\frac{{\bar{y}}_{r}^{b}}{y_{r0}^{b}})}$$
$$s.t.:\bar{x}\geq \sum_{j=1,\neq k}^{n}\theta_{j}x_{j};\;{\bar{y}}^{g}\leq \sum_{j=1,\neq k}^{n}\theta_{j}y_{j}^{g};\;{\bar{y}}^{b}\geq \sum_{j=1,\neq k}^{n}\theta_{j}y_{j}^{b};\;\bar{x}\geq x_{0},\;{\bar{y}}^{g}\leq y_{0}^{g},\;{\bar{y}}^{b}\geq y_{0}^{b},\;\theta \geq 0.$$
where $\rho^{*}$ denotes the efficiency value of the DMU, which can exceed 1. The definitions of the other variables are the same as those for Equation (4).

To more accurately assess the NCSE, and with reference to previous studies [9,10], three inputs were chosen, namely labor, capital, and energy. The GDP and CSV are desired indicators, and CO_2_ emissions are non-desired output indicators. The measurement results of these indicators are reported in Table 2.

(3)NCSE influencing factor data

According to the Introduction, the influencing factors of the NCSE may include the population density, industrial structure, per capita GDP, science and technology level, foreign investment, per capita road area, built-up area, road area, etc. These indicators are introduced in Table 3. The influencing factors were calculated based on statistical yearbooks across China from 2007 to 2019, including the China Statistical Yearbook, Beijing Statistical Yearbook, Tianjin Statistical Yearbook, Hebei Statistical Yearbook, Shanghai Statistical Yearbook, Jiangsu Statistical Yearbook, Zhejiang Statistical Yearbook, and Guangdong Statistical Yearbook.

#### 2.2.2. Spatiotemporal Cube Model (STC) Local Outlier Analysis

The STC is defined as a 3D visual form of geographic phenomena represented by the horizontal (spatial) and vertical (temporal) axes of the cube. It is a spatiotemporal model based on the aggregation of sample points, and can be used for geovisual analysis [55]. Each cube consists of attribute values corresponding to a specific period, with the start time at the bottom and the end time at the top, and the magnitude of these values can be distinguished by the use of different colors. In this study, the analysis was performed by calculating the NCSE values for the three UAs. Each cube represents the NCSE values for one city for one year, and multiple cubes were arranged vertically according to temporal order and aggregated into a spatiotemporal sequence. Statistically significant clusters and outliers in the spatiotemporal environment were calculated by examining the spatiotemporal statistical differences within the study area and its neighboring regions [56]. The neighborhood distance and neighborhood time step parameters were then used to estimate the Anselin Local Moran’s *I* statistics for each cubic bar column. A total of six tests were included: non-significance (the location was never statistically significant), high–high (H–H) clustering (the type of statistical significance of the position was always only H–H clustering), high–low (H–L) clustering (the type of statistical significance was always only H–L clustering), low–high (L–H) clustering (the type of statistical significance was always only L–H clustering), low–low (L–L) clustering (the type of statistical significance was always only L–L clustering), and multiple clustering (there were multiple statistically significant types of clustering and outlier types).

#### 2.2.3. Two-Way Fixed-Effects Model

The NCSE and impact factor data were combined with time-series and cross-sectional data to observe the changes in the impact factors in 38 cities within the three major UAs from 2007 to 2019. The expression is as follows.
(6)$$Y_{\mathrm{it}}=\alpha +X_{it}\beta_{i}+\varepsilon_{it}$$

After introducing each influencing factor into the model, the panel data model of this study were set as follows:(7)$$Y_{it}=\alpha +\beta_{1}\ln PD_{it}+\beta_{2}\ln S_{it}+\beta_{3}\ln PGDP_{it}+\beta_{4}\ln TEC_{it}+\beta_{5}\ln FDI_{it}+\beta_{6}\ln BU_{it}+\beta_{7}\ln PRO_{it}+\varepsilon_{it}$$
where *Y* is the explanatory variable denoting the NCSE. Moreover, *PD*, *INS*, *PGDP*, *TEC*, *FDI*, *BU*, and *PRO* are the explanatory variables (Table 3), which, respectively, denote the population density, industrial structure, GDP per capita, technology level, foreign investment, built-up area, and road area per capita. Finally, subscripts *i* and *t* denote the region and year, respectively; α denotes constant terms, $\beta_{i}$ denotes variable coefficients, and $\varepsilon_{it}$ denotes residuals.

After standardizing the data, regression analysis was carried out using this model, and F-tests and Hausman tests were performed in turn to determine whether there was an individual time effect after adding annual dummy variables. An endogeneity problem may arise from the mutual causality between the NCSE and each explanatory variable, and each explanatory variable may also have a time lag effect. Thus, each explanatory variable was taken separately with another lag period, and the robustness of the model was then further tested.

## 3. Analysis of the Results

### 3.1. Amount of Carbon Release

The carbon emissions of the three major UAs from 2007 to 2019 were calculated according to Equation (1). From Figure 2, it can be seen that the carbon emissions of Chengde in the BTH UA were the lowest and those of Tianjin were the highest. Except for Beijing, the carbon emissions of all the cities exhibited a fluctuating upward trend, among which Langfang, Tianjin, and Baoding had the largest emissions, with increases of more than 50%. The carbon emissions of 16 cities in the core of the YRD UA have also exhibited fluctuating upward trends over the past decade, with as many as 13 cities presenting a decreasing trend in carbon emissions in 2019 as compared to 2018. Shanghai was found to always be in the leading position in the YRD UA in terms of carbon emissions, while Zhoushan was found to have the lowest carbon emissions. The city with the highest carbon emissions in the PRD UA was Guangzhou, while Zhuhai was found to have the lowest. In addition, Huizhou was found to have the highest incremental carbon emissions, while Dongguan had the lowest.

### 3.2. Amount of Carbon Sequestration

The carbon sequestration volumes of the three major UAs from 2007 to 2019 were calculated based on Equation (2) and in combination with land use data. From Figure 3, it can be seen that the overall carbon sequestration volume of the BTH UA did not fluctuate greatly, and those of all the cities except Chengde presented a fluctuating downward trend but decreased less. Chengde was found to have the highest carbon sequestration volume, which increased by 16.48%. The carbon sequestration rates of all 16 cities in the YRD core area, except for Zhoushan, displayed a decreasing trend in the past 13 years, with the highest amount of carbon sequestration in Hangzhou and the lowest amount in Zhoushan. The overall carbon sequestration in the PRD UA was found to be less volatile, with all cities except Guangzhou showing a decreasing trend but with a smaller decrease. The carbon sequestration of Guangzhou increased by 1.6%, and that of Zhongshan decreased by 9.42%.

### 3.3. Analysis of the NCSE

The NCSE values of the three major UAs from 2007 to 2019 were calculated according to Equations (4) and (5). Figure 4 reveals that only four cities in the BTH UA presented an increase in NCSE in 2019 as compared to 2007, among which Tangshan and Langfang had the highest increases. In contrast, Beijing displayed the greatest decrease in NCSE as compared to 2007. Only Zhenjiang, Taizhou, and Shaoxing in the YRD UA presented decreases in NCSE in 2019 as compared to 2007. Shaoxing had the largest decrease, while Shanghai, Wuxi, and Suzhou had the largest increases. The NCSE values of Jiangmen, Zhaoqing, and Huizhou in the PRD UA were decreased in 2019 as compared to 2007, and that of Zhaoqing decreased the most. Shenzhen and Guangzhou were the two cities with the largest increases in NCSE in the PRD UA.

### 3.4. Analysis of NCSE Local Outliers

From Figure 5, it can be seen that the cities in the BTH UA with L–L clustering were mainly Beijing during 2009–2018 and Xingtai and Handan during 2015–2019, and the only city with H–H clustering was Chengde in 2017. Langfang presented multiple clustering patterns and spatiotemporal clustering during the entire study period. Nanjing, Suzhou, and their neighboring cities in the YRD UA showed a surge in NCSE values in 2019; these cities exhibited H–H spatiotemporal clustering and their neighboring cities presented L–H clustering. Shanghai and Hangzhou showed a multi-cluster pattern. Moreover, in the PRD UA, Guangzhou, Shenzhen, and Zhuhai presented a multi-cluster pattern, and Huizhou displayed H–H clustering in 2019. The remaining cities were not found to be spatially significant.

### 3.5. Analysis of the Influencing Factors of NCSE

The descriptive statistics of the influencing factors of NCSE are shown in Table 4. According to the method described in Section 2.2.3, the F-test was first used to determine whether the fixed-effects model or the mixed regression model should be adopted. The result was the rejection of the original hypothesis, i.e., the mixed regression model was found to be inapplicable to this study. Then, the Hausman test was chosen to determine whether a fixed-effects model (FE_robust) should be adopted instead of a random-effects model (RE). However, the traditional fixed-effects model only considered individual effects, so annual dummy variables were added to examine the presence of individual temporal effects [57]. After considering the temporal effect, the *p*-value of the joint significance test of all annual dummy variables was found to be less than 0.05, which strongly rejected the original hypothesis of “no temporal effect”. The two-way fixed-effects model was found to have a better fit than the fixed-effects model, and there was no multicollinearity among the influencing factors. Thus, the two-way fixed-effects model (FE_TW_DED) was ultimately chosen for the analysis.

The regression results of the two-way fixed-effects model show that the influencing factors of the NCSE of the three major UAs were found to be significantly different (Table 5). The BTH UA was mainly influenced by the GDP per capita and industrial structure; the YRD UA was mainly influenced by the industrial structure; the PRD UA was mainly influenced by the population density, industrial structure, and technology level. The one-period-lagged robustness test of the model demonstrated that the correlation coefficients of each influencing factor were basically consistent with the results of the previous empirical analysis, and the significance of only some influencing factors had changed.

## 4. Discussion

### 4.1. Analysis of the Carbon-Neutral Capacity of the Three Major UAs in China Based on NCSE

The state of carbon deficit poses a great threat to regional sustainable development [58] and is a key issue that China must improve to achieve the dual carbon target in the future. Studies have been conducted in the context of China’s singular UAs [40], typical provinces [59], and cities [60,61]. These studies found that all scales are characterized by carbon deficits caused by the decrease in carbon sequestration and the increase in carbon emissions, which is basically consistent with the findings of the present research. The results of this study show that the carbon emissions of the three major UAs all exhibited a fluctuating upward trend during the study period, while the amount of carbon sequestration fluctuated less, presenting a trend of first decreasing and then increasing. Overall, all three UAs were found to have a carbon deficit problem.

The NCSE is a measure of carbon balance by adding the GDP and CSV as expected outputs to the assessment framework with constant labor, capital, and energy inputs. Thus, the NCSE is more suitable for developing countries than measures used in previous studies, and is also more conducive to the sustainability of economic development.

In the exploration of the spatial distributions of the NCSE in the three major UAs in China using the STC model, different UAs were found to have different distribution characteristics. Beijing and its surrounding cities in the BTH UA, and Guangzhou, Shenzhen, and Zhuhai in the PRD UA, which are the more developed cities, were found to have a long, continuous period of exhibiting L–L clustering from the spatial perspective. In contrast, the YRD UA is completely different because it covers more provinces and cities, and the clustering pattern of Shanghai and Hangzhou was found to often change spatiotemporally in a multi-cluster pattern. The total economic volume of the YRD UA reached a new level in 2019 and became a national strategy in that year, which significantly increased the combined influence of various factors and caused Nanjing, Suzhou, and their surrounding cities to appear as H–H clusters in that year. In general, some cities in the three major UAs were found to have a common spatiotemporal clustering pattern. Moreover, the NCSE values of some cities fluctuated considerably, with more obvious fluctuations in the YRD UA. Although the carbon emissions of some cities are still increasing, the NCSE is also on an upward trend, which further confirms that simply relying on the reduction of carbon emissions and ignoring other factors to achieve carbon neutrality will hinder the sustainable economic development of developing countries.

### 4.2. Analysis of Influencing Factors

The causes of carbon neutrality in the three UAs were analyzed by using the NCSE as a measure of carbon neutrality. The BTH UA is more obviously influenced by government policy control than the other two UAs [62], and its relationship with the per capita GDP was found to be significant and positively correlated. When economic development reaches a certain level, residents and governments will pay more attention to improving environmental quality and promoting sustainable development [63]. With the increase in per capita income, to meet people’s demand for ecological environment quality, city governments will strengthen urban greening management and environmental regulation, promote the development of the urban low-carbon economy, and thus improve the urban NCSE.

In the YRD UA, the influence of the industrial structure was found to be significant and positively correlated with the NCSE. The upgrading of the industrial structure plays a positive role in the NCSE and contributes to the realization of carbon neutrality. City governments should actively promote industrial transfer, facilitate a reasonable industrial layout, and improve the energy use and pollution control efficiency via technological progress [51]. The YRD UA is diversified in terms of industrial categories, which mainly include electronic equipment manufacturing, metal smelting and processing, chemical material and product manufacturing, and the textile industry, as well as other light industries [64]. Therefore, the YRD is more significantly affected by the industrial structure as compared to the other two UAs.

The PRD UA was found to be significantly and negatively affected by the population density, which is consistent with previous findings indicating that urbanization has a negative impact on carbon sequestration [65]. Population agglomeration causes the rapid expansion of built-up areas; this places great pressure on urban energy consumption and the ecological environment, which has a negative impact on the urban NCSE. Shenzhen, with an urbanization rate close to 100%, was found to have the largest rate of change in NCSE among the cities in the PRD UA. In the process of urbanization, a large amount of arable forest land is converted into construction land, thus leading to a decrease in land carbon sequestration [66]. The high rate of urbanization and rapid economic development consume large amounts of resources and energy, which in turn directly leads to an increase in carbon emissions [67]. A more favorable factor is technological innovation, as it is crucial for the reduction of pollution emissions [13] and can effectively improve the energy use efficiency and energy mix conversion in cities. This, in turn, will reduce carbon emissions and is an important means by which to promote NCSE. Shenzhen and Guangzhou were found to have relatively high STI indices, which caused the factor of technological innovation to be significantly and positively influential in the PRD UA.

### 4.3. Policy Recommendations

In addition to enhancing the management of forest resources and expanding the scale of green spaces [68,69] to improve the carbon sequestration capacity of vegetation, the three UAs should adopt targeted improvement policies for the mitigation of the carbon deficit to promote a carbon balance according to the significance of the influencing factors. The BTH UA should increase investment, and the government should provide greener infrastructure and strongly support technological innovation [70]. It should also promote the popularity and use of new energy vehicles and increase subsidies. Regarding agriculture, low-carbon agricultural technology should be developed and the value of food and agricultural waste should be increased [71]. The YRD UA should continue to take advantage of its industrial structure to develop clean energy, such as wind, water, solar, biomass, hydrogen, thermal, ocean, nuclear, and new material energy storage, according to local conditions [72]. The carbon trading capacity of enterprises in industrial production should be strengthened [73], the elimination of high-energy-consuming industries should be promoted, and industrial upgrading and adjustment should be promoted [74]. To improve the NCSE of the cities, the PRD UA should actively consider the role of foreign investors and utilize their unique advantages to further enhance the capacity of science and technology innovation through domestic and foreign trade and technology exchange [21].

## 5. Conclusions

The three major UAs in China were selected as the research objects of this study, and their carbon sequestration and carbon release from energy consumption were respectively calculated according to the LUCCs during 2007–2019. The NCSE values of the three major UAs were then calculated by the non-expectation super-efficiency SBM model, and their spatiotemporal evolution characteristics were analyzed based on the STC model. The findings of this study reveal the following. (1) The carbon sequestration in the three major UAs was found to show a fluctuating trend of generally decreasing, but the interannual variation was small. The carbon release mostly showed a fluctuating upward trend and a deficit level in general, which cannot achieve a carbon balance. In the BTH UA, the NCSE values of Tangshan, Chengde, and Langfang were higher than those of Beijing and Tianjin. In the YRD UA, the NCSE values of Shanghai and its neighboring cities were higher than those of Zhenjiang and Changzhou. In the PRD UA, the NCSE values of Guangzhou and Shenzhen were higher than those of Foshan, Dongguan, Zhongshan, and Zhuhai. (2) The cities in the L–L clusters in the BTH UA were mainly Beijing and its surrounding cities during 2009–2018 and Xingtai and Handan from 2015 to 2019. The only city with H–H clustering was Chengde in 2017. The NCSE values in the YRD UA centered on Shanghai, Nanjing, and their surrounding cities revealed H–H clustering in 2019, while Changzhou, Ningbo, and their surrounding cities presented L–H clustering. The NCSE values of the remaining cities in the PRD UA, namely Guangzhou, Shenzhen, and Zhuhai, showed a multi-cluster pattern that was not spatiotemporally significant, and the spatiotemporal clusters were scattered. (3) Among the three major UAs, the BTH UA was found to be influenced by and positively correlated with the industrial structure and GDP per capita; the YRD UA was significantly influenced by and positively correlated with the industrial structure; the PRD UA was influenced by the population density, industrial structure, and technology level, with the former being negatively correlated and the latter two being positively correlated.

This study analyzed the NCSE distributions of the three major UAs in China at the spatial and temporal scales using the STC model in the context of sustainable economic development. This undertaking was different from previous analyses targeting individual time points, and the research method was more scientific. The NCSE was calculated by combining carbon emissions and CSV with constant labor, capital, and energy inputs, which can better reflect the comprehensive ability of CO_2_ reduction and carbon sequestration. Unlike the traditional single-variable analysis, the carbon-neutral status of three major UAs in China over more than ten consecutive years was discussed in terms of carbon emissions, CSV, and NCSE. Under different NCSE levels, the carbon-neutral policies of each region should be formulated according to the significance of the influencing factors in the local context. In addition, this study was characterized by some shortcomings. In the econometric model testing, a lag of only one period was used to test the robustness and endogeneity of the model. Moreover, because the scope of the study involved data from different provincial and municipal statistical yearbooks over several years, there were missing data for some years, and thus the final results may have some errors. In the future, more advanced models and indicators should be considered and selected to measure the carbon balance, and the influencing factors should be studied from the perspective of counties in a more refined manner. In conclusion, China’s three major UAs have a long way to go to achieve carbon neutrality, and as representative UAs in a developing country, they should play a leading role as models for other urbanized regions in developing countries to formulate carbon-neutral policies in the context of balancing development and the environment.

## Figures and Tables

**Figure 1 ijerph-20-01166-f001:**
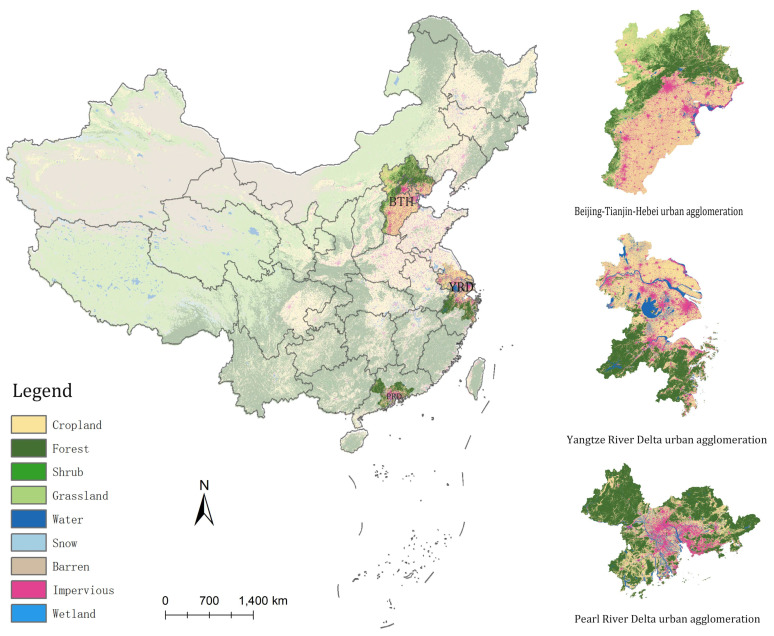
The location and LUCC map of the three major UAs in China.

**Figure 2 ijerph-20-01166-f002:**
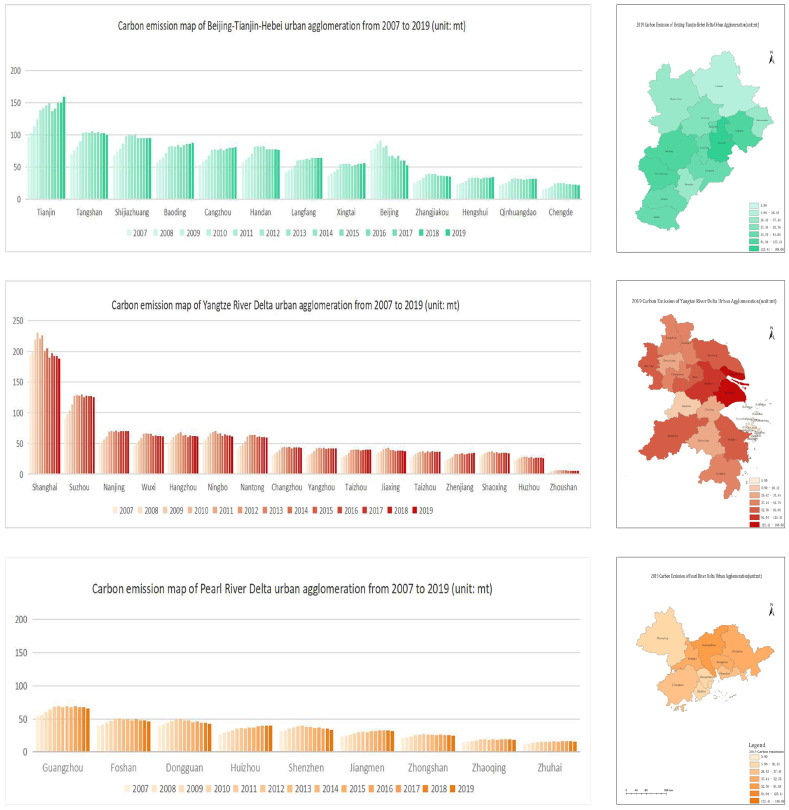
The carbon emissions of the three major UAs from 2007 to 2019.

**Figure 3 ijerph-20-01166-f003:**
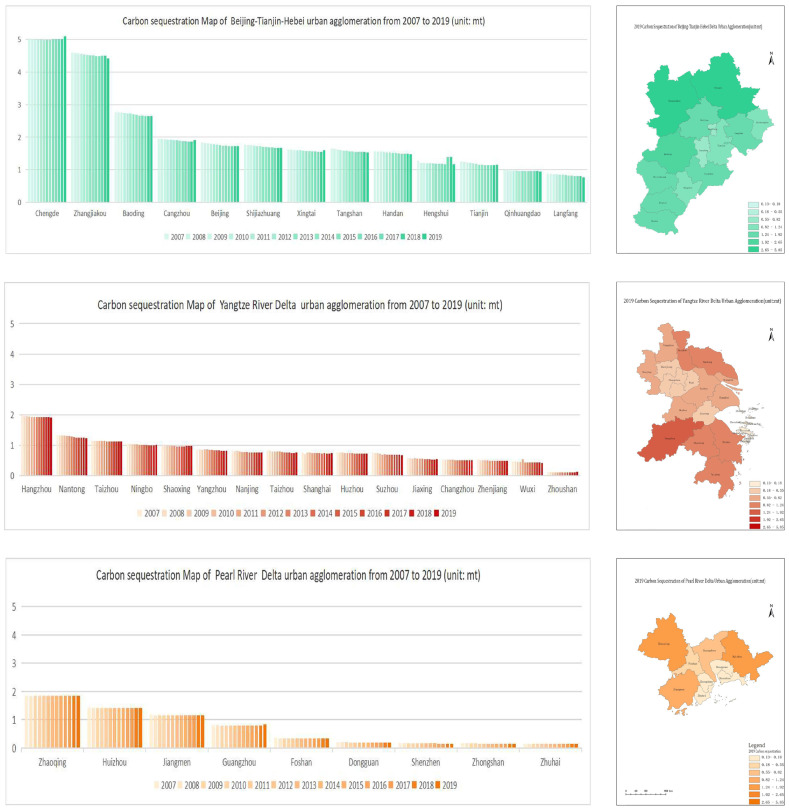
The carbon sequestration of the three major UAs from 2007 to 2019.

**Figure 4 ijerph-20-01166-f004:**
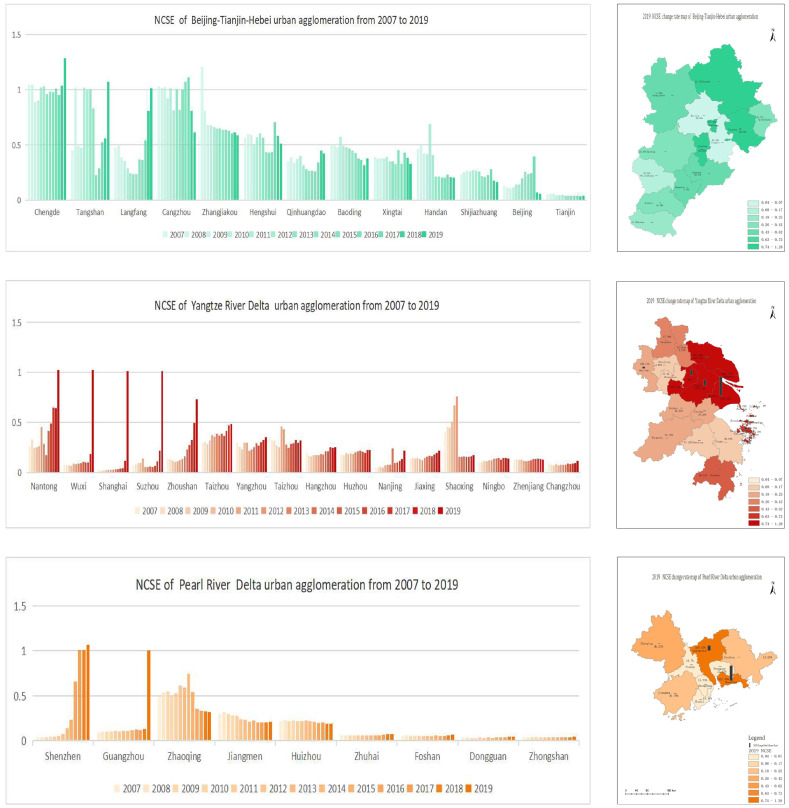
The NCSE maps of the three major UAs from 2007 to 2019.

**Figure 5 ijerph-20-01166-f005:**
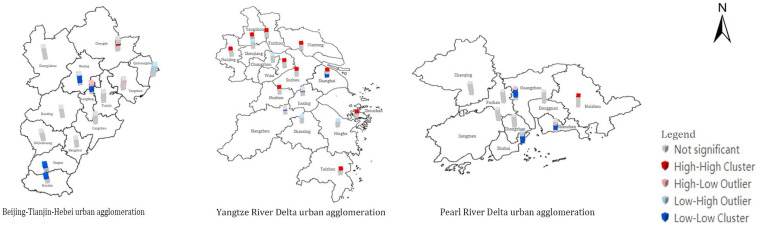
The NCSE STC diagrams of the three major UAs from 2007 to 2019.

**Table 1 ijerph-20-01166-t001:** The carbon sequestration coefficients based on the LUCC type dataset.

LUCC Type	Unit	Carbon Sequestration Factor (*δ*)	Source
Grassland	t/hm^2^	0.95	[43]
Cropland	t/hm^2^	1.7	[44]
Shrub	t/hm^2^	0.134	[45]
Barren	t/hm^2^	0.005	[46]
Forest	t/hm^2^	1.21	[47]
Water	t/hm^2^	0.24	[46]

**Table 2 ijerph-20-01166-t002:** The input–output index system.

Variable Type Representation and Unit	Measurement Index	Representation and Unit
Input	Capital input	Investment stock of fixed assets (million RMB)
	Labor input	Employed workers (ten thousand people)
	Energy consumption	Coal consumption (ten thousand tons)
Desirable output	GDP	Annual real GDP (billion RMB)
	CSV	Carbon sequestration of vegetation (million tons)
Undesirable output	CO_2_ emissions	Energy consumption CO_2_ emissions (million tons)

**Table 3 ijerph-20-01166-t003:** The introduction of the variables.

Variable Type	Variable	Meaning and Unit of Variable	Calculation Method	Possibility	Reference
Dependent variable	NCSE	Net carbon sink efficiency (%)	See Section 2.2.1		[20]
Independent variable	PD	Population density (people/km^2^)	Ln (Number of population/area)	Population congregation will cause the built-up area to expand rapidly, placing enormous pressure on urban energy consumption and the ecological environment (PD → −NCSE)	[15,16]
	INS	Industrial structure (%)	Ln (Tertiary industry output value/secondary industry output value)	Industrial upgrading and adjustment help to eliminate high-energy-consumption industries (INS → +NCSE)	[50,51]
	PGDP	GDP per capita (RMB)	Ln (Total GDP/total population)	Economically developed countries pay more attention to environmental protection and thus invest more in emission reduction (PGDP → +NCSE)	[37,52]
	TEC	Technological progress (pieces)	Ln (Number of patents in the current year)	Technological progress helps to improve efficiency and promote the replacement of energy-intensive industries (TEC → +NCSE)	[13]
	FDI	Foreign direct investment (ten thousand USD)	Ln (Amount of foreign investment)	Foreign investment may bring advanced technology (FDI → +NCSE)	[21]
	PRO	Per road area (m^2^)	Ln (Urban road area/urban non-agricultural population)	The increase in the road area eases traffic congestion and reduces urban CO_2_ emissions (PRO → +NCSE)	[53]
	BU	Built-up area (km^2^)	Ln (Area of built-up area)	Urban sprawl occupies green areas and reduces the carbon sequestration capacity of vegetation (BU → −NCSE)	[54]

**Table 4 ijerph-20-01166-t004:** The descriptive statistics of the influencing factors of the NCSE of the three major UAs.

	Variable	Obs	Mean	Std. Dev.	Min	Max
BTH	NCSE	169	0.483	0.308	0.036	1.282
	PD	169	7.881	0.664	6.492	9.383
	INS	169	1.159	0.935	0.413	5.168
	PGDP	169	10.583	0.584	9.45	12.01
	TEC	169	7.609	1.581	4.762	11.788
	FDI	169	11.067	1.519	7.012	14.941
	BU	169	5.013	0.966	3.758	7.292
	PRO	169	15.924	3.731	5.26	27.949
TRD	NCSE	208	0.213	0.176	0.023	1.026
	PD	208	7.634	0.494	6.524	8.545
	INS	208	0.984	0.384	0.561	2.695
	PGDP	208	11.301	0.425	10.186	12.1
	TEC	208	9.469	1.19	5.1	11.519
	FDI	208	12.144	1.127	8.813	14.46
	BU	208	5.273	0.818	3.895	7.121
	PRO	208	20.467	6.611	4.04	34.21
PRD	NCSE	117	0.185	0.222	0.032	1.063
	PD	117	7.867	0.51	6.468	8.976
	INS	117	0.992	0.421	0.513	2.631
	PGDP	117	11.307	0.569	9.674	13.056
	TEC	117	9.187	1.377	5.525	12.023
	FDI	117	12.06	0.815	9.806	13.617
	BU	117	5.557	0.954	4.218	7.189
	PRO	117	16.016	5.447	8.28	33.47

**Table 5 ijerph-20-01166-t005:** The estimation results and robustness tests of the two-way fixed-effects model for the three major UAs in China.

	BTH	YRD	PRD	BTH One-Period Lagged	YRD One-Period Lagged	PRD One-Period Lagged
PD	0.107	0.082	−0.139 ***	0.162	0.039	−0.106
INS	0.185 **	0.178 ***	0.195 **	0.277 *	0.229 ***	0.385 *
PGDP	0.268 **	0.243	0.002	0.114 *	0.142	−0.108 **
TEC	0.027	−0.047	0.108 **	−0.097	−0.055 *	0.144
FDI	−0.006	−0.003	0.154	−0.014	-0.006	0.111
BU	−0.14	−0.284	-0.367	−0.184	−0.227	−0.442
PRO	−0.001	0.012	-0.005	0.001	0.020 **	0.000
R^2^	0.219	0.391	0.306	0.158	0.219	0.301

Note: * *p* < 0.05, ** *p* < 0.01, *** *p* < 0.001.

## Data Availability

The data presented in this study are openly available from FigShare at https://doi.org/10.6084/m9.figshare.21782210.v1.

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
