# Peer review of "Analysis of the Spatiotemporal Evolution of the Net Carbon Sink Efficiency and Its Influencing Factors at the City Level in Three Major Urban Agglomerations in China"

_ijerph, 2023, doi:10.3390/ijerph20021166_

Round 1

Reviewer 1 Report

The overall writing of this paper is rigorous and the demonstration process is standardized.

However, there are too many such articles, which lack originality and are of little significance.

The theory of the paper is not enough.

Author Response

Dear reviewer,

Thank you for your comments about the revision of our manuscript. The current calculation of indicators to measure the carbon balance based on the land-use context mostly focuses on a single indicator, such as the analysis of the impact factors of carbon emission alone or the analysis of the impact of CSV. Few measurements integrate multiple indicators, such as carbon emission and carbon sequestration, to form a new standard. In this study, we used the net carbon sink efficiency (NCSE) to measure the carbon balance. Only one previous study has used this indicator to calculate the spatial and temporal distributions of the NCSE in different regions of China and to analyze the influencing factors. In addition, we have subsequently listed some studies that measured the carbon balance in different regions. It is evident that there is not yet a precise indicator for the carbon balance in the international arena, which provides a meaningful research consideration.

Title

Index

Study area

Spatial-temporal evolution and influencing factors of
 net carbon sink efficiency in Chinese cities under the background of carbon neutrality

Net carbon
sink efficiency(NCSE)

China

Spatiotemporal Analysis of Carbon Emissions and
Carbon Storage Using National Geography Census
Data in Wuhan, China

Carbon emissions
 and carbon storage (CECS)

Wuhan, China

Carbon sources/sinks analysis of land use changes in China based on data envelopment
 analysis  

 Carbon emission
efficiency(CEE)

China

Regional carbon fluxes from land use and land cover change in Asia,
1980–2009

 Carbon budget

Asia

Effects of contemporary land-use and land-cover change
on the carbon balance of terrestrial ecosystems in the
United States

Net carbon sink

United States

China is a vast country with differences among its regions, and a unified normative guide is lacking. In this research, as an application-oriented study, we found that different regions are driven by different influencing factors by studying the typical three major urban agglomerations in China. Thus, the realization for each region cannot be measured with the same label, so it is meaningful and necessary to calculate the NCSE.

Reviewer 2 Report

This is an interesting study and its quality is good, I think. However, I would like to see the following improvements in the manuscript before making a concrete decision:

1. the authors should provide a separate and extended literature review section.

2. the authors should indicate the shortcomings of the adopted econometric methodology.

3. the authors should provide a footnote that they are willing to share their data set in Excel format with those who wish to replicate the results of this research.

4. the authors should present some guidelines for future studies.

5. Is the income nominal or real?

Author Response

Dear reviewer,

Thank you for your comments about the revision of this manuscript. The current research literature on the NCSE is scarce and cannot be combed for a systematic literature review. Nevertheless, the scales and influencing factors considered in the extant research have been summarized and expanded in the Introduction section. The shortcomings regarding econometric studies are mainly reflected in the robustness and endogeneity tests of the model using only one-period lags, and this has also been supplemented in the Conclusion. The data used for this research have been made publicly available on FigShare. Guidelines for future research have also been added to the Conclusion, mainly in terms of the possibility of studying the spatial distribution of the NCSE, the more refined analysis of influencing factors from a county perspective, and the selection of more precise measurement modalities and data to measure the carbon balance. Finally, the income issue is the real income, mainly from the National Key R&D Program of China for some research data purchase and collection and related model learning expenses.

Reviewer 3 Report

Shen et al. used the net carbon sink efficiency (NCSE) to measure and analyze three of China’s major urban area carbon emissions and sequestration capacity. They also highlight the critical factors impacting these areas’ carbon budget. I think this is an interesting work. It has the potential to help the policy design and development plan for a large urban area. It is publishable if several issues can be addressed. The English also needs to be improved.

1.     Abstract. The author should rewrite the results part in the abstract. First, you need to explain the different clustering meanings, such as H-H and L-H. And you need to use their full name before the abbreviation. Try to extract short takeaway messages from your results; your three results are vague and distracting.

2.     Line 37: “from people from all walks of life” is redundant; delete it.

3.      Lines 55-line 66, this part could be more precise. It is hard to have a clear understanding of these terminologies. Please give clear and easy-follow descriptions. NCSE in line 64 is the first time introduced, but there is no full name. 

4.     Line 174, NCSE calculation. It is very hard to follow and understand.

5.     Table 3. See 1.2.1. I need help finding 1.2.

6.     Line 237, explaining the different clustering patterns and their implications is better.

7.     Discussion, there are assumptions and uncertainties in this study. You could discuss these uncertainties.

Author Response

Dear reviewer,

Thank you for your comments on the revision of our manuscript. The English language of the manuscript has been re-touched and revised. The abbreviations of the different clusters in the Abstract have also been expanded as full terms. Any deletions and additions to the full names that needed to be made have been improved. The descriptions of the CEE and CSV in lines 55-66 of the paper have been reorganized, and the meanings of the different clustering patterns have been explained and clarified. The assumptions and uncertainties about this study have also been discussed in the Conclusion.

Round 2

Reviewer 2 Report

The revised manuscript reads fine

Author Response

Dear reviewer

Thank you for your comments about the revision of this manuscript.  

Reviewer 3 Report

The manuscript is improved in many aspects. I would like to see a more detailed and point-by-point response.

Thank you.

Author Response

Dear reviewer,

Thank you for your comments about the revision of this manuscript.  

  1. The meaning of each cluster is explained systematically in Section 2.2.2, and the full names of the clusters are provided in the Abstract. The first point summarizes the overall carbon sequestration and emissions of the three major urban agglomerations.Itthen outlines the clustering of the NCSE values of each urban agglomeration in time and space, as well as the factors that respectively affect them.
  2. People from various industries have been removed.
  3. The relationship between CEE and CSV has beenreorganized and added in lines 55-66.To consider CEE and CSV together, NCSE has been introduced as an indicator by which to measure the carbon balance standard. The full term of the acronym NCSE has also been added.
  4. Line 174 mainly concerns the calculation method of CSV: the area of different land-use types is multiplied by the carbon sequestration coefficient. The calculation of NCSE mainly includes the input indicators of the capital stock, labor force,and energy,the expected output indicators of CSV and GDP, and the non-expected output indicator of carbon emissions. The amount of carbon sequestered by vegetation is the expected output indicator in the calculation of NCSE.
  5. Thishas been changed; see Section 1.2.
  6. Themeaning of each clusterhas been explained in detail.
  7. The uncertainties and assumptions inthis researchmainly concern the possible shortcomings of the robustness of the econometric model and the lack of certain errors in the statistical yearbook data used. These factors contribute to the uncertainty of this study, and are discussed in the Conclusion section.
  8. We invited English professionals to polish the article.